# The Multi-Agent Behavior Dataset: Mouse Dyadic Social Interactions

**Jennifer J. Sun**
Caltech

**Tomomi Karigo**
Caltech

**Dipam Chakraborty**
AICrowd Research

**Sharada P. Mohanty**
AICrowd Research

**Benjamin Wild**
Freie Universität Berlin

**Quan Sun**
OPPO Research Institute

**Chen Chen**
OPPO Research Institute

**David J. Anderson**
Caltech

**Pietro Perona**
Caltech

**Yisong Yue**
Caltech

**Ann Kennedy**
Northwestern University
`ann.kennedy@northwestern.edu`

**Dataset Website:** `https://sites.google.com/view/computational-behavior/our-datasets/calms21-dataset`

## Abstract

Multi-agent behavior modeling aims to understand the interactions that occur between agents. We present a multi-agent dataset from behavioral neuroscience, the Caltech Mouse Social Interactions (CalMS21) Dataset. Our dataset consists of trajectory data of social interactions, recorded from videos of freely behaving mice in a standard resident-intruder assay. To help accelerate behavioral studies, the CalMS21 dataset provides benchmarks to evaluate the performance of automated behavior classification methods in three settings: (1) for training on large behavioral datasets all annotated by a single annotator, (2) for style transfer to learn inter-annotator differences in behavior definitions, and (3) for learning of new behaviors of interest given limited training data. The dataset consists of 6 million frames of unlabeled tracked poses of interacting mice, as well as over 1 million frames with tracked poses and corresponding frame-level behavior annotations. The challenge of our dataset is to be able to classify behaviors accurately using both labeled and unlabeled tracking data, as well as being able to generalize to new settings.

## 1 Introduction

The behavior of intelligent agents is often shaped by interactions with other agents and the environment. As a result, models of multi-agent behavior are of interest in diverse domains, including neuroscience [55], video games [26], sports analytics [74], and autonomous vehicles [8]. Here, we study multi-agent animal behavior from neuroscience and introduce a dataset to benchmark behavior model performance.

Traditionally, the study of animal behavior relied on the manual, frame-by-frame annotation of behavioral videos by trained human experts. This is a costly and time-consuming process, and cannot easily be crowdsourced due to the training required to identify many behaviors accurately. Automated behavior classification is a popular emerging tool [29, 2, 18, 44, 55], as it promises to reduce human annotation effort, and opens the field to more high-throughput screening of animal behaviors. However, there are few large-scale publicly available datasets for training and benchmarking social behavior classification, and the behaviors annotated in those datasets may not match the set of behaviors a particular researcher wants to study. Collecting and labeling enough training data to

reliably identify a behavior of interest remains a major bottleneck in the application of automated analyses to behavioral datasets.

We present a dataset of behavior annotations and tracked poses from pairs of socially interacting mice, the **Cal**tech **M**ouse **S**ocial Interactions 2021 (CalMS21) Dataset, with the goal of advancing the state-of-the-art in behavior classification. From top-view recorded videos of mouse interactions, we detect seven keypoints for each mouse in each frame using Mouse Action Recognition System (MARS) [55]. Accompanying the pose data, we introduce three tasks pertaining to the classification of frame-level social behavior exhibited by the mice, with frame-by-frame manual annotations of the behaviors of interest (Figure 1), and additionally release video data for a subset of the tasks. Finally, we release a large dataset of tracked poses without behavior annotations, that can be used to study unsupervised learning methods.

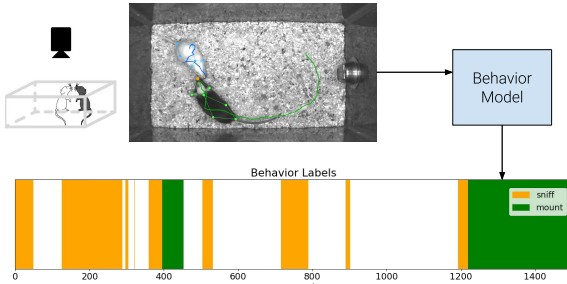

Figure 1: **Overview of behavior classification.** A typical behavior study starts with extraction of tracking data from videos. We show 7 keypoints for each mouse, and draw the trajectory of the nose keypoint. The goal of the model is to classify each frame (30Hz) to one of the behaviors of interest from domain experts.

As part of the initial benchmarking of CalMS21, we both evaluated standard baseline methods as well as solicited novel methods by having CalMS21 as part of the Multi-Agent Behavior (MABe) Challenge 2021 hosted at CVPR 2021. To test model generalization, our dataset contains splits annotated by different annotators and for different behaviors.

In addition to providing a performance benchmark for multi-agent behavior classification, our dataset is suitable for studying several research questions, including: How do we train models that transfer well to new conditions (annotators and behavior labels)? How do we train accurate models to identify rare behaviors? How can we best leverage unlabeled data for behavior classification?

## 2 Related Work

**Behavior Classification.** Automated behavior classification tools are becoming increasingly adopted in neuroscience [55, 19, 29, 2, 44]. These automated classifiers generally consists of the following steps: pose estimation, feature computation, and behavior classification. Our dataset provides the output from our mouse pose tracker, MARS [55], to allow participants in our dataset challenge to focus on developing methods for the latter steps of feature computation and behavior classification. We will therefore first focus our exploration of related works on these two topics, specifically within the domain of neuroscience, then discuss how our work connects to the related field of human action recognition.

Existing behavior classification methods are typically trained using tracked poses or hand-designed features in a fully-supervised fashion with human-annotated behaviors [27, 55, 5, 19, 44]. Pose representations used for behavior classification can take the form of anatomically defined keypoints [55, 44], fit shapes such as ellipses [27, 12, 46], or simply a point reflecting the location of an animal's centroid [45, 51, 68]. Features extracted from poses may reflect values such as animal velocity and acceleration, distances between pairs of body parts or animals, distances to objects or parts of the arena, and angles or angular velocities of keypoint-defined joints. To bypass the effort-intensive step of hand-designing pose features, self-supervised methods for feature extraction have been explored [59]. Computational approaches to behavior analysis in neuroscience have been recently reviewed in [50, 13, 17, 2].

Relating to behavior classification and works in behavioral neuroscience, there is also the field of human action recognition (reviewed in [71, 75]). We compare this area to our work in terms of models and data. Many works in action recognition are trained end-to-end from image or video data [38, 57, 63, 7], and the sub-area that is most related to our dataset is pose/skeleton-based action recognition [10, 37, 60], where model input is also pose data. However, one difference is that these

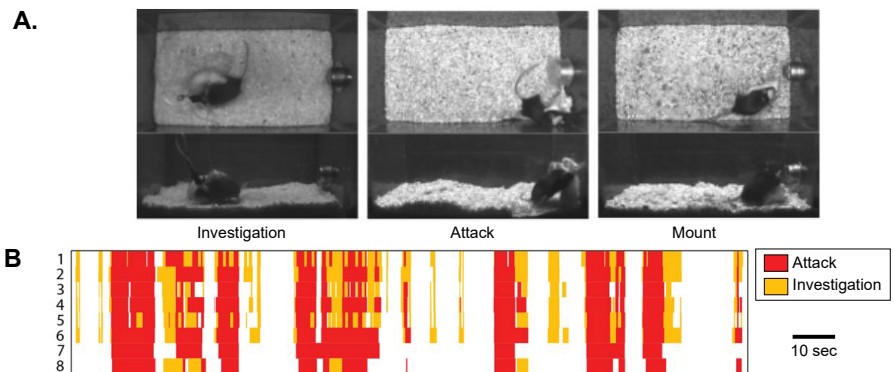

Figure 2: **Behavior classes and annotator variability.** A. Example frames showing some behaviors of interest. B. Domain expert variability in behavior annotation, reproduced with permission from [55]. Each row shows annotations from a different domain expert annotating the same video data.

models often aim to predict one action label per video, since in many datasets, the labels are annotated at the video or clip level [7, 76, 56]. More closely related to our dataset is the works on temporal segmentation [34, 58, 33], where one action label is predicted per frame in long videos. These works are based on human activities, often goal-directed in a specific context, such as cooking in the kitchen. We would like to note a few unique aspects animal behavior. In contrast to many human action recognition datasets, naturalistic animal behavior often requires domain expertise to annotate, making it more difficult to obtain. Additionally, most applications of animal behavior recognition are in scientific studies in a laboratory context, meaning that the environment is under close experimenter control.

**Unsupervised Learning for Behavior.** As an alternative to supervised behavior classification, several groups have used unsupervised methods to identify actions from videos or pose estimates of freely behaving animals [3, 32, 69, 64, 39, 28, 41] (also reviewed in [13, 50]). In most unsupervised approaches, videos of interacting animals are first processed to remove behavior-irrelevant features such as the absolute location of the animal; this may be done by registering the animal to a template or extract a pose estimate. Features extracted from the processed videos or poses are then clustered into groups, often using a model that takes into account the temporal structure of animal trajectories, such as a set of wavelet transforms [3], an autoregressive hidden Markov model [69], or a recurrent neural network [39]. Behavior clusters produced from unsupervised analysis have been shown to be sufficiently sensitive to distinguish between animals of different species and strains [25, 39], and to detect effects of pharmacological perturbations [70]. Clusters identified in unsupervised analysis can often be related back to human-defined behaviors via post-hoc labeling [69, 3, 64], suggesting that cluster identities could serve as a low-dimensional input to a supervised behavior classifier.

**Related Datasets.** The CalMS21 dataset provides a benchmark to evaluate the performance of behavior analysis models. Related animal social behavior datasets include CRIM13 [5] and Fly vs. Fly [19], which focus on supervised behavior classification. In comparison to existing datasets, CalMS21 enables evaluation in multiple settings, such as for annotation style transfer and for learning new behaviors. The trajectory data provided by the MARS tracker [55] (seven keypoints per mouse) in our dataset also provides a richer description of the agents compared to single keypoints (CRIM13). Additionally, CalMS21 is a good testbed for unsupervised and self-supervised models, given its inclusion of a large (6 million frame) unlabeled dataset.

While our task is behavior classification, we would like note that there are also a few datasets focusing on the related task of multi-animal tracking [48, 22]. Multi-animal tracking can be difficult due to occlusion and identity tracking over long timescales. In our work, we used the output of the MARS tracker [55], which also includes a multi-animal tracking dataset on the two mice to evaluate pose tracking performance; we bypass the problem of identity tracking by using animals of differing coat colors. Improved methods to more accurately track multi-animal data is another direction that can help quantify animal behavior.

Other datasets studying multi-agent behavior include those from autonomous driving [8, 61], sports analytics [73, 14], and video games [54, 23]. Generally, the autonomous vehicle datasets focus on tracking and forecasting, whereas trajectory data is already provided in CalMS21, and our focus is on behavior classification. Sports analytics datasets also often involves forecasting and learning player strategies. Video game datasets have perfect tracking and generally focus on reinforcement learning or imitation learning of agents in the simulated environment. While the trajectories in CalMS21 can be used for imitation learning of mouse behavior, our dataset also consist of expert human annotations of behaviors of interest used in scientific experiments. As a result, CalMS21 can be used to benchmark supervised or unsupervised behavior models against expert human annotations of behavior.

## 3    Dataset Design

The CalMS21 dataset is designed for studying behavior classification, where the goal is to assign frame-wise labels of animal behavior to temporal pose tracking data. The tracking data is a top-view pose estimate of a pair of interacting laboratory mice, produced from raw 30Hz videos using MARS [55], and reflecting the location of the nose, ears, neck, hips, and tail base of each animal (Figure 3).

We define three behavior classification tasks on our dataset. In Task 1 (Section 3.1), we evaluate the ability of models to classify three social behaviors of interest (attack, mount, and close investigation) given a large training set of annotated videos; sample frames of the three behaviors are shown in Figure 2A. In Task 2 (Section 3.2), models must be adjusted to reproduce new annotation styles for the behaviors studied in Task 1: Figure 2B demonstrates the annotator variability that can exist for the same videos with the same labels. Finally, in Task 3 (Section 3.3), models must be trained to classify new social behaviors of interest given limited training data.

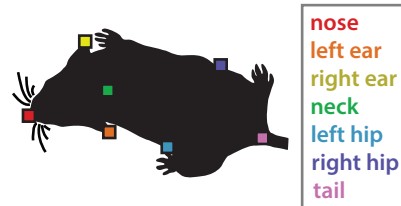

Figure 3: **Pose keypoint definitions.** Illustration of the seven anatomically defined keypoints tracked on the body of each animal. Pose estimation is performed using MARS [55].

In Tasks 1 and 2, each frame is assigned one label (including "other" when no social behavior is shown), therefore these tasks can be handled as multi-class classification problems. In Task 3, we provide separate training sets for each of seven novel behaviors of interest, where in each training set only a single behavior has been annotated. For this task, model performance is evaluated for each behavior separately: therefore, Task 3 should be treated as a set of binary classification problems. Behaviors are temporal by nature, and often cannot be accurately identified from the poses of animals in a single frame of video. Thus, all three tasks can be seen as time series prediction problems or sequence-to-sequence learning, where the time-evolving trajectories of 28-dimensional animal pose data (7 keypoints x 2 mice x 2 dimensions) must be mapped to a behavior label for each frame. Tasks 2 and 3 are also examples of few-shot learning problems, and would benefit from creative forms of data augmentation, task-efficient feature extraction, or unsupervised clustering to stretch the utility of the small training sets provided.

To encourage the combination of supervised and unsupervised methods, we provide a large set of unlabeled videos (around 6 million frames) that can be used for feature learning or clustering in any task (Figure 4).

### 3.1    Task 1: Classical Classification

Task 1 is a standard sequential classification task: given a large training set comprised of pose trajectories and frame-level annotations of freely interacting mice, we would like to produce a model to predict frame-level annotations from pose trajectories on a separate test set of videos. There are 70 videos in the public training set, all of which have been annotated for three social behaviors of interest: close investigation, attack, and mount (described in more detail in the appendix). The goal is for the model to reproduce the behavior annotation style from the training set.

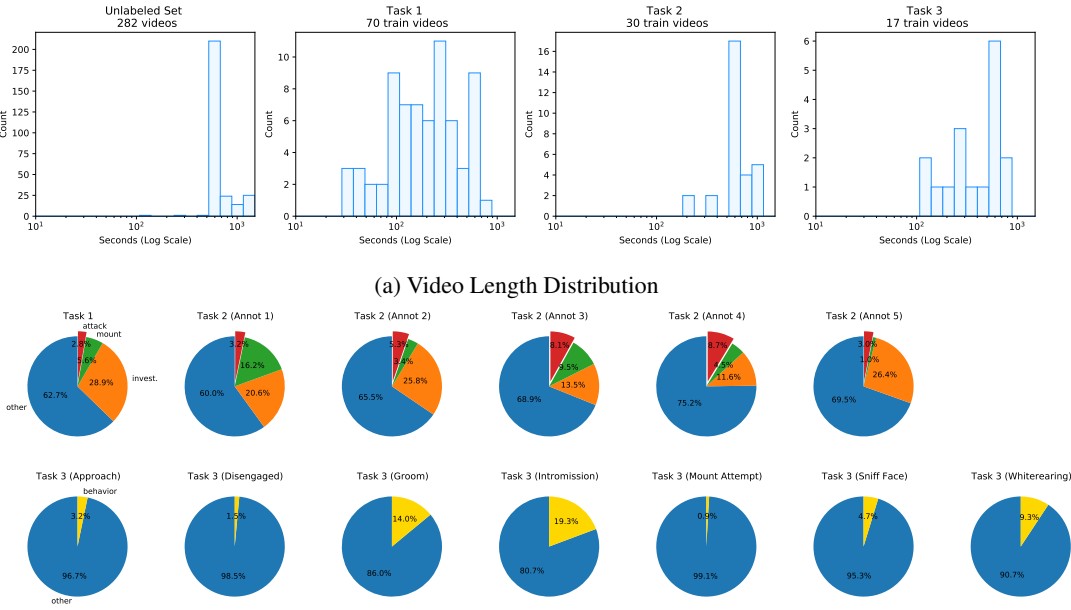

(a) Video Length Distribution

(b) Percentage of Annotated Behaviors

Figure 4: **Available data for each task in our challenge.** Our dataset consists of a large set of unlabeled videos alongside a set of annotated videos from one annotator. Annot 1, 2, 3, 4, 5 are different domain experts, whose annotations for attack, mount, and investigation are used in Task 2. Bottom row shows new behaviors used in Task 3.

Sequential classification has been widely studied, existing works use models such as recurrent neural networks [11], temporal convolutional networks [35], and random forests with hand-designed input features [55]. Input features to the model can also be learned with self-supervision [59, 9, 36], which can improve classification performance using the unlabeled portion of the dataset.

In addition to pose data, we also release all source videos for Task 1, to facilitate development of methods that require visual data.

### 3.2 Task 2: Annotation Style Transfer

In general, when annotating the same videos for the same behaviors, there exists variability across annotators (Figure 2B). As a result, models that are trained for one annotator may not generalize well to other annotators. Given a small amount of data from several annotators, we would like to study how well a model can be trained to reproduce each individual's annotation style. Such an "annotation style transfer" method could help us better understand differences in the way behaviors are defined across annotators and labs, increasing the reproducibility of experimental results. A better understanding of different annotation styles could also enable crowdsourced labels from non-experts to be transferred to the style of expert labels.

In this sequential classification task, we provide six 10-minute training videos for each of five annotators unseen in Task 1, and evaluate the ability of models to produce annotations in each annotator's style. All annotators are trained annotators from the David Anderson Lab, and have between several months to several years of prior annotation experience. The behaviors in the training datasets are the same as Task 1. In addition to the annotator-specific videos, competitors have access to a) the large collection of task 1 videos, that have been annotated for the same behaviors but in a different style, and b) the pool of unannotated videos, which could be used for unsupervised clustering or feature learning.

This task is suitable for studying techniques from transfer learning [62] and domain adaptation [65]. We have a source domain with labels from task 1, which needs to be transferred to each annotator in task 2 with comparatively fewer labels. Potential directions include learning a common set

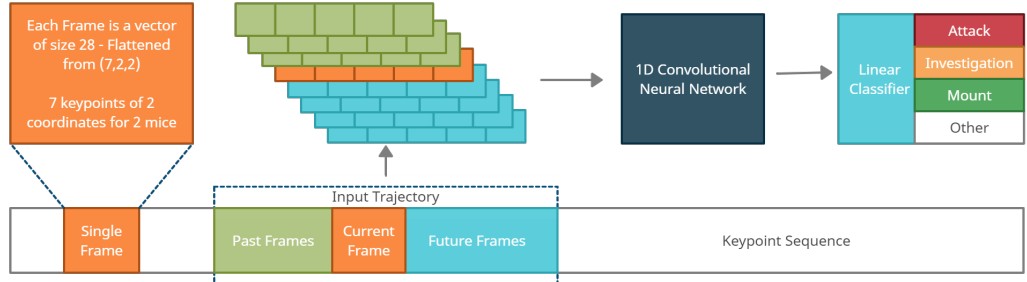

Figure 5: **Sequence Classification Setup.** Sequence information from past, present, and future frames may be used to predict the observed behavior label on the current frame. Here, we show a 1D convolutional neural network, but in general any model may be used.

of data-efficient features for both tasks [59], and knowledge transfer from a teacher to a student network [1].

### 3.3 Task 3: New Behaviors

It is often the case that different researchers will want to study different behaviors in the same experimental setting. The goal of Task 3 is to help benchmark general approaches for training new behavior classifiers given a small amount of data. This task contains annotations on seven behaviors not labeled in Tasks 1 & 2, where some behaviors are very rare (Figure 4).

As for the previous two tasks, we provide a training set of videos in which behaviors have been annotated on a frame-by-frame basis, and evaluate the ability of models to produce frame-wise behavior classifications on a held-out test set. We expect that the large unlabeled video dataset (Figure 4) will help improve performance on this task, by enabling unsupervised feature extraction or clustering of animal pose representations prior to classifier training.

Since each new behavior has a small amount of data, few-show learning techniques [66] can be helpful for this task. The data from Task 1 and the unlabeled set could also be used to set up multi-task learning [77], and for knowledge transfer [1].

## 4 Benchmarks on CalMS21

We develop an initial benchmark on CalMS21 based on standard baseline methods for sequence classification. To demonstrate the utility of the unlabeled data, we also used these sequences to train a representation learning framework (task programming [59]) and added the learned trajectory features to our baseline models. Additionally, we presented CalMS21 at the MABe Challenge hosted at CVPR 2021, and we include results on the top performing methods for each of the tasks.

Our evaluation metrics are based on class-averaged F1 score and Mean Average Precision (more details in the appendix). Unless otherwise stated, the class with the highest predicted probability in each frame was used to compute F1 score.

### 4.1 Baseline Model Architectures

Our goal is to estimate the behavior labels in each frame from trajectory data, and we use information from both past and future frames for this task (Figure 5). To establish baseline classification performance on CalMS21, we explored a family of neural network-based classification approaches (Figure 6). All models were trained using categorical cross entropy loss [21] on Task 1: Classic Classification, using an 80/20 split of the train split into training and validation sets during development. We report results on the full training set after fixing model hyperparameters.

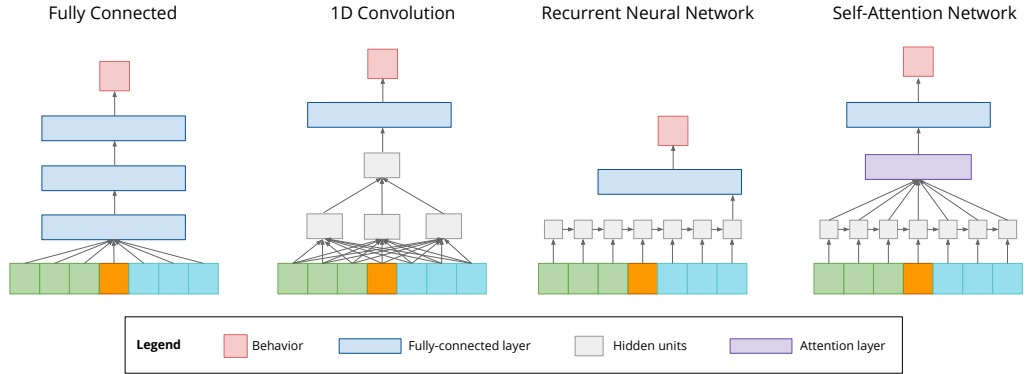

Figure 6: **Baseline models.** Different baseline setups we evaluated for behavior classification. The input frame coloring follows the same convention as Figure 5: past frames in green, current frame in orange, and future frames in cyan.

Among the explored architectures, we obtained the highest performance using the 1D convolutional neural network (Table 1). We therefore used this architecture for baseline evaluations in all subsequent sections.

Hyperparameters we considered includes the number of input frames, the numer of frame skips, the number of units per layer, and the learning rate. Settings of these parameters may be found in the project code and the appendix. The baseline with task programming models use the same hyperparameters as the baseline. The task programming model is trained on the unlabeled set only, and the learned features are concatenated with the keypoints at each frame.

### 4.2 Task 1 Classic Classification Results

**Baseline Models.** We used the 1D convolutional neural network model outlined above (Figure 5) to predict attack, investigation, and mount behaviors in two settings: using raw pose trajectories, and using trajectories plus features learned from the unlabeled set using task programming (Table 2). We found that including task programming features improved model performance. Many prediction errors of the baseline models were localized around behavior transition boundaries (Figure 7). These errors may arise in part from annotator noise in the human generated labels of behaviors. An analysis of such intra- (and inter-) annotator variability is found in [55].

| Method | Average F1 | MAP |
|---|---|---|
| Fully Connected | $.659 \pm .005$ | $.726 \pm .004$ |
| LSTM | $.675 \pm .011$ | $.712 \pm .013$ |
| Self-Attention | $.610 \pm .028$ | $.644 \pm .018$ |
| 1D Conv Net | $.793 \pm .011$ | $.856 \pm .010$ |

Table 1: Class-averaged results on Task 1 (attack, investigation, mount) for different baseline model architectures. The value is shown of the mean and standard deviation over 5 runs.

**Task1 Top-1 Entry.** We also include the top-1 entry in Task 1 of our dataset at MABe 2021 as part of our benchmark (Table 2). This model starts from an egocentric representation of the data; in a preprocessing stage, features are computed based on distances, velocities, and angles between coordinates relative to the agents' orientations. Furthermore, a PCA embedding of pairwise distances of all coordinates of both individuals is given as input to the model.

The model architecture is based on [47] with three main components. First, the embedder network consists of several residual blocks [24] of non-causal 1D convolutional layers. Next, the contexter network is a stack of residual blocks with causal 1D convolutional layers. Finally, a fully connected residual block with multiple linear classification heads computes the class probabilities for each behavior. Additional inputs such as a learned embedding for the annotator (see Section 4.3) and absolute spatial and temporal information are directly fed into this final component.

The Task 1 top-1 model was trained in a semi-supervised fashion, using the normalized temperature-scaled cross-entropy loss [47, 9] for all samples and the categorical cross-entropy loss for labeled samples. During training, sequences were sampled from the data proportional to their length with a 3:1 split of unlabeled/labeled sequences. Linear transformations that project the contexter component's

| Method | Data Used During Training | | | Average F1 | MAP |
|---|---|---|---|---|---|
| | Task 1 (train split) | Unlabeled Set | All Tasks (all splits) | | |
| Baseline | ✓ | | | $.793 \pm .011$ | $.856 \pm .010$ |
| Baseline w/ task prog | ✓ | ✓ | | $.829 \pm .004$ | $.889 \pm .004$ |
| MABe 2021 Task 1 Top-1 | ✓ | ✓ | ✓ | $.864 \pm .011$ | $.914 \pm .009$ |

Table 2: Class-averaged results on Task 1 (attack, investigation, mount; mean $\pm$ standard deviation over 5 runs.) See appendix for per class results. The "All Tasks" column indicates that the model was jointly trained on all three tasks, and "all splits" indicates that both labeled train set and trajectory from unlabeled test set are used.

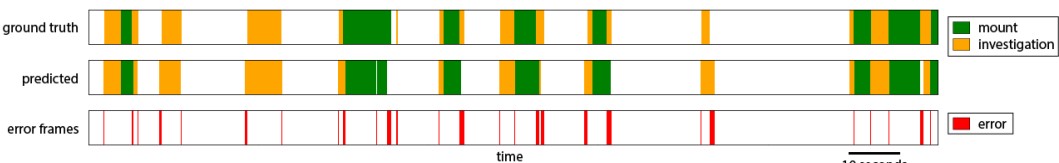

Figure 7: Example of errors from a sequence of behaviors from Task 1.

outputs into the future are learned jointly with the model, as described in [47]. This unsupervised loss component regularizes the model by encouraging the model to learn a representation that is predictive of future behavior. A single model was trained jointly for all three tasks of the challenge, with all parameters being shared among the tasks, except for the final linear classification layers. The validation loss was monitored during training, and a copy of the parameters with the lowest validation loss was stored for each task individually.

### 4.3 Task 2 Annotation Style Transfer Results

**Baseline Model.** Similar to Task 1, Task 2 involves classifying attack, investigation, and mount frames. However, in this task, our goal is to capture the particular annotation style of different individuals. This step is important in identifying sources of discrepancy in behavior definitions between datasets or labs. Given the limited training set size in Task 2 (only 6 videos for each annotator), we used the model trained on Task 1 as a pre-trained model for the baseline experiments in Task 2, to leverage the larger training set from Task 1. The performances are in Table 3, with per-annotator results in the appendix.

**Task2 Top-1 Entry.** The top MABe submission for Task 2 re-used the model architecture and training schema from Task 1, described in Section 4.2. To address different annotation styles in Task 2, a learned annotator embedding was concatenated to the outputs of the contexter network. This embedding was initialized as a diagonal matrix such that initially, each annotator is represented by a one-hot vector. This annotator matrix is learnable, so the network can learn to represent similarities in the annotation styles. Annotators 3 and 4 were found to be similar to each other, and different from annotators 1, 2, and 5. The learned annotator matrix is provided in the appendix.

### 4.4 Task 3 New Behaviors Results

**Baseline Model.** Task 3 is a set of data-limited binary classification problems with previously unseen behaviors. Although these behaviors do occur in the Task 1 and Task 2 datasets, they are not labeled. The challenges in this task arise from both the low amount of training data for each new behavior and the high class imbalance, as seen in Figure 4.

For this task, we used our trained Task 1 baseline model as a starting point. Due to the small size of the training set, we found that models that did not account for class imbalance performed poorly. We therefore addressed class imbalance in our baseline model by replacing our original loss function with a weighted cross-entropy loss in which we scaled the weight of the under-represented class by the number of training frames for that class. Results for Task 3 are provided in Table 4. We found classifier performance to depend both on the percentage of frames during which a behavior was observed, and on the average duration of a behavior bout, with shorter bouts having lower classifier performance.

| Method | Data Used During Training | | | Average F1 | MAP |
|---|---|---|---|---|---|
| | Task 1 & 2 (train split) | Unlabeled Set | All Tasks (all splits) | | |
| Baseline | ✓ | | | .754 ± .005 | .813 ± .003 |
| Baseline w/ task prog | ✓ | ✓ | | .774 ± .006 | .835 ± .005 |
| MABe 2021 Task 2 Top-1 | ✓ | ✓ | ✓ | .809 ± .015 | .857 ± .007 |

Table 3: Class-averaged and annotator-averaged results on Task 2 (attack, investigation, mount; mean ± standard deviation over 5 runs). The "All Tasks" column indicates that the model was jointly trained on all three tasks, and "all splits" indicates that both labeled train set and trajectory from unlabeled test set are used. See appendix for per class and per annotator results.

| Method | Data Used During Training | | | Average F1 | MAP |
|---|---|---|---|---|---|
| | Task 1 (train split) | Task 3 (train split) | Unlabeled Set | | |
| Baseline | ✓ | ✓ | | 0.338 ± .004 | .317 ± .005 |
| Baseline w/ task prog | ✓ | ✓ | ✓ | .328 ± .009 | .320 ± .009 |
| MABe 2021 Task 3 Top-1 | | ✓ | | .319 ± .025 (.363 ± .020) | .352 ± .023 |

Table 4: Class-averaged results on Task 3 over the 7 behaviors of interest (mean ± standard deviation over 5 runs.) The average F1 score in brackets corresponds to improvements with threshold tuning. See appendix for per class results.

**Task3 Top-1 Entry.** The model for the top entry in Task 3 of the MABe Challenge was inspired by spatial-temporal graphs that have been used for skeleton-based action recognition algorithms in human pose datasets. In particular, MS-G3D [37] is an effective algorithm for extracting multi-scale spatial-temporal features and long-range dependencies. The MS-G3D model is composed of a stack of multiple spatial-temporal graph convolution blocks, followed by a global average pooling layer and a softmax classifier [37].

The spatial graph for Task 3 is constructed using the detected pose keypoints, with a connection added between the necks of the two mice. The inputs are normalized following [72]. MS-G3D is then trained in a fully supervised fashion on the train split of Task 3. Additionally, the model is trained with data augmentation based on rotation.

## 5 Discussion

We introduce CalMS21, a new dataset for detecting the actions of freely behaving mice engaged in naturalistic social interactions in a laboratory setting. The released data include over 70 hours of tracked poses from pairs of mice, and over 10 hours of manual, frame-by-frame annotation of animals' actions. Our dataset provides a new way to benchmark the performance of multi-agent behavior classifiers. In addition to reducing human effort, automated behavior classification can lead to more objective, precise, and scalable measurements compared to manual annotation [2, 16]. Furthermore, techniques studied on our dataset can be potentially applied to other multi-agent datasets, such as those for sports analytics and autonomous vehicles.

In addition to the overall goal of supervised behavior classification, we emphasize two specific problems where we see a need for further investigation. The first of these is the utility of behavior classifiers for comparison of annotation style between different individuals or labs, most closely relating to our Task 2 on annotation style transfer. The ability to identify sources of inter-annotator disagreement is important for the reproducibility of behavioral results, and we hope that this dataset will foster further investigation into the variability of human-defined behavior annotations. A second problem of interest is the automated detection of new behaviors of interest given limited training data. This is especially important for the field of automated behavior analysis, as few-shot training of behavior classifiers would enable researchers to use supervised behavior classification as a tool to rapidly explore and curate large datasets of behavioral videos.

Alongside manually annotated trajectories provided for classifier training and testing, we include a large set of unlabeled trajectory data from 282 videos. The unlabeled dataset may be used to improve the performance of supervised classifiers, for example by learning self-supervised representations of trajectories [59], or it may be used on its own for the development of unsupervised methods for behavior discovery or trajectory forecasting. We note that trajectory forecasting is a task that is of interest to other fields studying multi-agent behavior, including self-driving cars and sports analytics. We hope that our dataset can provide an additional domain with which to test these models. In addition, unsupervised behavior analysis may be capable of identifying a greater number of behaviors than a human annotator would be able to annotate reliably. Recent single-animal work has shown that unsupervised pose analyses can enable the detection of subtle differences between experimental conditions [70]. A common problem in unsupervised analysis is evaluating the quality of the learned representation. Therefore, an important topic to be addressed in future work is the development of appropriate challenge tasks to evaluate the quality of unsupervised representations of animal movements and actions, beyond comparison with human-defined behaviors.

**Broader Impact.** In recent years, animal behavior analysis has emerged as a powerful tool in the fields of biology and neuroscience, enabling high-throughput behavioral screening in hundreds of hours of behavioral video [4]. Prior to the emergence of these tools, behavior analysis relied on manual frame-by-frame annotation of animals' actions, a process which is costly, subjective, and arduous for domain experts. The increased throughput enabled by automation of behavior analysis has seen applications in neural circuit mapping [53, 6], computational drug development [70], evolution [25], ecology [16], and studies of diseases and disorders [40, 27, 70]. In releasing this dataset, our hope is to establish community benchmarks and metrics for the evaluation of new computational behavior analysis tools, particularly for social behaviors, which are particularly challenging to investigate due to their heterogeneity and complexity.

In addition to behavioral neuroscience, behavior modeling is of interest to diverse fields, including autonomous vehicles, healthcare, and video games. While behavior modeling can help accelerate scientific experiments and lead to useful applications, some applications of these models to human datasets, such as for profiling users or for conducting surveillance, may require more careful consideration. Ultimately, users of behavior models need to be aware of potentially negative societal impacts caused by their application.

**Future Directions.** In this dataset release, we have opted to emphasize behavior classification from keypoint-based animal pose estimates. However, it is possible that video data could further improve classifier performance. Since we have also released accompanying video data to a subset of CalMS21, an interesting future direction would be to determine the circumstances under which video data can improve behavior classification. Additionally, our dataset currently focuses on top-view tracked poses from a pair of interacting mice. For future iterations, including additional organisms, experimental settings, and task designs could further help benchmark the performance of behavior classification models. Finally, we value any input from the community on CalMS21 and you can reach us at mabe.workshop@gmail.com.

# 6   Acknowledgements

We would like to thank the researchers at the David Anderson Research Group at Caltech for this collaboration and the recording and annotation of the mouse behavior datasets, in particular, Tomomi Karigo, Mikaya Kim, Jung-sook Chang, Xiaolin Da, and Robert Robertson. We are grateful to the team at AICrowd for the support and hosting of our dataset challenge, as well as Northwestern University and Amazon Sagemaker for funding our challenge prizes. This work was generously supported by the Simons Collaboration on the Global Brain grant 543025 (to PP), NIH Award #K99MH117264 (to AK), NSF Award #1918839 (to YY), NSERC Award #PGSD3-532647-2019 (to JJS), as well as a gift from Charles and Lily Trimble (to PP).

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
