# OpenReview forum: "The Multi-Agent Behavior Dataset: Mouse Dyadic Social Interactions"
_NeurIPS.cc/2021/Track/Datasets_and_Benchmarks/Round1 — NeurIPS 2021 Datasets and Benchmarks Track (Round 1)_

### Official Review · Reviewer_NFev · 2021-07-01
**Well constructed multi-animal action recognition dataset**

**Rating:** 8
**Confidence:** 5
**Clarity:** The manuscript is very clear.

**Strengths:**

## Relevance
Animal behavioral tracking is becoming an established field, drawing tremendous enthusiasm from multiple fields. This includes neuroscience, which has long been a subject at NeurIPS. Algorithm choices for behavioral identification, as well as other tasks, are often ad hoc in the field, and a move towards standardizing datasets for specific challenges makes sense. Social behavioral experiments are becoming more and more common in neuroscience in particular, making this dataset of interest to a growing community.

## Significance
This work was first used as a challenge competition and I think that it has already demonstrated excellence in that regard. I think the majority of users will use it to iterate techniques for action recognition in animal models, although it is possible that  some will want to analyze the pose coordinates outside of the action labels.

## Accessibility and accountability
The manuscript is well written, the documentation of methods complete, and has already been used to organize a challenge competition at CVPR2021 with dozens of participants.



**Weaknesses:**

This is framed as a multi-agent behavior dataset, but in many ways, it is an action recognition dataset, and it is somewhat limited in that respect, mainly focusing on 3 actions, although 10 are given in Task 3. It is possible that this could be used for more multi-agent style problems, for instance predicting the impact individuals have on one another’s trajectories, but this is not explored.

 Much of the novelty in the dataset provided is in the keypoint tracking, as a portion of the authors published a similar dataset of dyadic rodent interactions in 2012 (CRIM13, Burgos-Artizzu et al.; cited), with I believe additional behaviors labeled. I do think that the dataset is valuable, as in many experiments scientists are only interested in a portion of behaviors animals perform. However I am not sure it will find a general usage as a multi-animal action recognition dataset due to the limited number of actions provided. I also am not sure about the utility of the annotator style transfer task.

There could be more discussion of the relevant human action recognition literature.


**Additional Feedback:**

I thank the authors for an excellent submission. I would enjoy hearing if there were plans to provide continuous action labels, and whether the animal pose tracks themselves would be useful for other applications. Are there other applications that this dataset could find utility beyond the three challenges described?

**Correctness:**

I believe the manuscript is sound and that the claims are well documented. The benchmarks, having already been used for a challenge, are excellent.

**Documentation:**

The work is well documented.

**Ethics:**

No ethical considerations.

**Relation To Prior Work:**

Some prior work on multi-animal datasets is discussed, although I do feel like the scope of discussion is somewhat narrow. Investigators in the multi-animal keypoint tracking field are beginning to release datasets, which could be cited. While articles in diverse field (driving, sports) are mentioned, I am not sure how relevant they are to the specific challenges solved in this dataset, as they are used for very different tasks. I think references to the extensive human action recognition literature would be more apt. I was left wondering how similar the SoA benchmarks achieved were to contemporary approaches for human action recognition, or if there are specific challenges for working with animals and in particular multiple animals.

**Summary And Contributions:**

Update: July 20, 2021: Thank you for the responses to my comments and adding additional information about pose tracking approaches. Overall I think the related work is clearer and improved. I stand by the original score and vote to accept.


Animal behavioral analysis is of growing interest to many disciplines, but there are a dearth of datasets for benchmarking algorithms for behavioral recognition and other tasks. This contribution describes the CalMS21 dataset, which consists of 1 million video frames of tracked animal poses and behavioral annotations for mice interacting in a resident-intruder assay. The dataset is primarily used for action recognition tasks with 3 and 10 actions, but also contains annotations in some cases from multiple annotators, making it useful for ‘style transfer’ across individuals. The dataset is thoroughly documented, and freely available. Baselines for performance on three different tasks are presented, and moreover, the dataset has already been used for a challenge competition at CVPR2021. The presented challenge solutions are interesting, and may be more broadly useful in the animal behavioral tracking field.

---

> ### Author Response · Authors · 2021-07-12
> **Response to Initial Review**
>
> Thanks for your thoughts and review! In our challenge and task definitions, we’ve focused on the supervised learning aspect of our data, but we want to point out that Section 5 addresses some of the further usage we see with our dataset. Section 5 also explains why we believe more people should investigate human annotator variability, and Task 2 is one step that we take towards this direction using automated behavior classifiers. We strongly agree that the dataset could be used for predicting the impact of individuals on each others’ trajectories, and for other multi-agent behavior problems. For future plans of our dataset, we are preparing tasks for future years to specifically highlight unsupervised learning + forecasting tasks rather than classification. We will expand on Section 5 with these points.
>
> We will add a paragraph discussing human action recognition in Section 2 (such as 29, A, B, etc.) with part of the extra page we are given. Some of our challenge participants indeed used models from human action recognition, such as MS-G3D [29] (task 3 top performing model). We will also add a paragraph on multi-animal tracking datasets, such as [C, D].
>
> Some specific challenges to working with animal behaviors are that animal behavior often requires domain expertise to annotate, and domain experts may study specific, new behaviors in different scientific studies. In contrast, often human behaviors/actions can more easily be annotated by non-experts and crowdsourced. Additionally, for animal behavior, there are many different species with different skeletons, whereas human datasets could be more easily pooled to take advantage of a similar skeleton structure and similar actions. An additional challenge of working with multiple animals is that we need to take into account interaction between animals - this aspect is more general to multi-agent behavior modeling.
>
> Please let us know if you have any other feedback, thanks! The updated manuscript revision will be up in a few days.
>
> [A]  Choutas, Vasileios, et al. "Potion: Pose motion representation for action recognition." Proceedings of the IEEE Conference on Computer Vision and Pattern Recognition. 2018.
>
> [B] Carreira, Joao, and Andrew Zisserman. "Quo vadis, action recognition? a new model and the kinetics dataset." proceedings of the IEEE Conference on Computer Vision and Pattern Recognition. 2017.
>
> [C] Pedersen, M., Haurum, J.B., Bengtson, S.H. & Moeslund, T.B. 3D-ZeF: A 3D Zebrafish Tracking Benchmark Dataset. CVPR2020.
>
> [D] Graving, Jacob M., et al. "DeepPoseKit, a software toolkit for fast and robust animal pose estimation using deep learning." Elife 8 (2019): e47994.

---

### Official Review · Reviewer_1vhc · 2021-07-03
**Initial review on Mouse Dyadic Social Interaction Dataset**

**Rating:** 6
**Confidence:** 4

**Strengths:**

This dataset will help investigation of an interesting problem domain through dyadic mouse social interaction and has relevance to other research fields in behavior modeling and classification. Moreover, the dataset is designed as a benchmark dataset on different tasks (fully supervised learning and few-shot learning) which is well motivated by the need for detecting rare and novel behavior in behavior research.
In terms of size, this dataset is quite large. While the majority of the videos in the dataset is unlabeled, they have value for studying unsupervised feature learning to facilitate future research.
Reports on baseline and top performing models from the MABe Challenge are informative and well documented.

**Weaknesses:**

There are number of questions raised around the Task 2 proposed in this paper, especially regarding [L160-161] “in this sequential classification task, we provide six 10-minute training videos for each of five new annotators, and evaluate the ability of models to produce annotations in each annotator’s style” [L167-168] “We have a source domain with labels from task 1, which needs to be transferred to each annotator in task 2 with comparatively fewer labels”.
Typically, human manual coding in behavioral study involves training of the human annotators such that they get reliability among themselves (inter-rater reliabilities are measured and then repeatedly trained until it reaches certain threshold). As this paper describes, novice annotators annotated labels for Task 2. Did these new annotators go through this kind of reliability training? In fact, it is never mentioned in the paper how the human annotation process differs between Task 1 and Task 2. In both Task 1 and Task 2, the label set is the same, then how come there is annotator bias in Task 2 whereas there isn’t in Task 1? If you train a machine learning model to mimic one novice annotator’s bias based on Task 2’s label, how do you tell whether the learned style reflects the untrained/unreliable annotation state VS true environmental bias that may exist in in certain lab? Please address these issues in the revision.

**Additional Feedback:**

Please see above comments.

**Clarity:**

In general the paper is easy to read and well written.
Perhaps the term “multi agent” repeatedly emphasized at the beginning of the paper is a bit of a stretch as this dataset is limited to dyadic behavior only and might need to be toned down.

**Correctness:**

Please refer to my comments in the weakness section above as I think the main weakness of the paper relates to the correctness/validity of the authors claim.


**Documentation:**

Details on dataset is available through the URL provided in the submission.
Please provide details on the number of annotators and their reliability across task 1, 2, 3 in the main article.

**Ethics:**

As the authors put [L314-317] “while behavior modeling can help accelerate scientific experiments and lead to useful applications, some applications of these models to human datasets, such as for profiling users or for conducting surveillance, may require more careful consideration”, there are potential risks when all behavior modeling research considered together, but the consequence from this particular paper is fairly limited as it only deals with mouse behavior in controlled lab environment.


**Relation To Prior Work:**

Key distinction of this work in comparison to prior work on animal social behavior (e.g. CRIM13 [4] and Fly vs Fly [14]) lies on the task that the dataset is designed to benchmark. While prior work only focused on supervised behavior classification, the proposed work enables evaluation in new behavior under un supervised/semisupervised context as it introduces a large number of unlabeled videos and includes few-shot labels for a subset of behaviors.

**Summary And Contributions:**

This paper presents a video dataset consisting of pairs of socially interacting mice with 6 million frames of unlabeled tracked poses and over 1 million frames with tracked poses and corresponding frame-level behavior annotations. Tracked poses are output from their pose tracker MARS [43]. Behavior annotations are designed for three different tasks; 1) classical closed-set supervised learning task. 2) task of mimicking one annotator’s annotation "style". 3) prediction of a rare behavior label given only a few shot training samples. Using this new dataset as benchmark dataset, the authors introduce baseline methods as well as a few top performing methods from the CVPR 2021 MABe challenge that the authors hosted recently.

---

> ### Author Response · Authors · 2021-07-12
> **Response to Initial Review**
>
> Thanks for your feedback. All human annotators for our behavior annotations are trained annotators from the David Anderson Lab (one annotator for Task 1, five annotators for Task 2, one annotator per behavior for Task 3) and have between several months to several years of prior annotation experience. We therefore do not consider them “novice” annotators, and we will clarify Section 4.2 so this is more clear- we only meant that these are previously unseen annotators relative to the annotations in Task 1.
>
> A note on inter-annotator variability: when new annotators in the lab are trained, they are provided with a description of each behavior they are to annotate, and supervised by a current lab member until consistency is established. Nonetheless, there are some remaining disagreements between annotators even within the lab, and likely there are even greater disagreements between different neuroscience labs studying the same behavior, as there are no “consensus” definitions of behaviors that are accurate enough to cover all cases in which the behavior might arise. In many experiments (e.g., a study measuring total aggression time expressed in different mouse strains), inter-annotator variability can be tolerated as an additional source of noise in the data. In other experiments where annotation accuracy is more critical (e.g., a study identifying the neural correlates of attack initiation), the same video will often be re-scored by multiple individuals until a consensus set of annotations is achieved. There is a more in depth exploration of annotator bias in animal behavioral studies in MARS, Segalin et al., including quantification of both inter-annotator and intra-annotator variability on a common set of videos. We include a figure from this study in a private comment to the reviewers.
>
> We designed Task 2 with the goal of establishing methods that could be used to compare annotation styles between labs, so that it would be possible to identify points where two labs may differ in their definitions of behavior. We hope that this will help improve reproducibility in behavioral neuroscience by helping labs compare their annotation datasets. We will clarify our writing in the main paper so this is more clear.
>
> Please let us know if you have any other feedback, thanks! The updated manuscript revision will be up in a few days.

---

### Official Review · Reviewer_uKmm · 2021-07-20
**Large-scale dataset for animal multi-agent behavior dataset**

**Rating:** 7
**Confidence:** 5
**Clarity:** The paper is well written.

**Strengths:**

The dataset contains a large number of annotated frames (1 million) with tracked poses and behavior labels and 6 million unlabelled frames with tracked poses.

Three tasks are proposed along with the dataset. The second task poses a challenge of transfer learning to a different annotation style, this direction is promising for the type of problems that require expert human annotators. The third task requires understanding novel behaviors with limited data. Proposed tasks on the dataset are useful for real-world applications and the dataset provided will facilitate the research.

The dataset was, to some degree, adopted by the community in the recent Multi-Agent Behavior (MABe) Challenge 2021.

**Weaknesses:**

The dataset has an application to a very narrow problem of understanding the behavior of mice in a very controlled environment. The methods developed on top of the provided annotations will rely on accurate pose predictions, in a real-world application the pose detections might not be as precise

**Additional Feedback:**

The authors could consider writing about the direct applications of the research this dataset will facilitate.


**Correctness:**

The claims are correct, evaluation methods are the winning methods of the challenge.



**Documentation:**

The data collection process could be explained in more detail, the authors don't explicitly comment on the maintenance plans for the dataset hosting.

The dataset website is clear and easy to follow. There is a well-documented code repository with the baseline codes.

**Ethics:**

I don't believe there are ethical concerns for the dataset.



**Relation To Prior Work:**

The prior work is clearly discussed.

**Summary And Contributions:**

Modified 19. July 2021: Thank you for the rebuttal and for addressing my questions. The dataset is a solid contribution to the community.
I am confident in my rating and vote to accept this paper.

The paper presents a dataset of mouse interactions, provided annotations include 7 keypoints of the mouse obtained using a pose estimation model. Along with the data, the authors propose three evaluation settings and the motivation behind them. The benchmark models proposed are the winning methods of the Multi-Agent Behavior Workshop.

---

### Author Response · Authors · 2021-07-14
**Updated Changes to Main Paper**

Thanks to everyone for your feedback! We've updated the pdf of our main paper with the following changes:

> New paragraphs in Section 2 (starting at line 76 and line 111)

We added new paragraphs to our Related Works section discussing human action recognition, and other multi-agent tracking datasets.

> Clarifications to Task 2 in Section 3.2 (starting at line 184)

We clarified the annotator experience for Task 2 in the main paper.

> Updated Section 5 and broader impact subsection.

We updated Section 5 with a reference to Task 2 in line 319, and clarified other applications of our dataset in line 330. We also added a paragraph in line 340 on broader impacts of animal behavior analysis.

> Updated task programming results so that random seed is consistent with baseline.

We updated the task programming results using the same random seeds as baseline. We did not observe a significant change.

Please let us know if you have any other comments, thanks again!

---

### Decision · Program_Chairs · 2021-07-26

**Decision:**

Accept

**Comment:**

The paper presents a dataset of 1M videos frames with tracked animal poses and behavior annotations. All reviewers agreed that the paper is well-written and that the dataset is useful -- as evidenced by the fact that it has already been used to organize a challenge. Reviewers raised some minor concerns including more details about annotators and the relationship between the proposed benchmark and prior work on human action recognition. The author response was satisfactory in addressing these concerns, and in the end all reviewers voted to accept the paper. Congratulations on having your paper accepted to the NeurIPS 2021 Track on Datasets and Benchmarks! The authors are encouraged to take the feedback from reviewers into account when preparing the camera-ready version of the paper.